# The Impact of Increased Fib-4 Score in Patients with Type II Diabetes Mellitus on COVID-19 Disease Prognosis

**DOI:** 10.3390/medicina57050434

**Published:** 2021-04-30

**Authors:** Ovidiu P. Calapod, Andreea M. Marin, Minodora Onisai, Laura C. Tribus, Corina S. Pop, Carmen Fierbinteanu-Braticevici

**Affiliations:** 1Gastroenterology Department, Emergency University Hospital of Bucharest, “Carol Davila” University of Medicine and Pharmacy, 050098 Bucharest, Romania; ovidiucalapod@yahoo.com (O.P.C.); marindeea29@gmail.com (A.M.M.); cfierbinteanu@yahoo.com (C.F.-B.); 2Hematology Department, Emergency University Hospital of Bucharest, “Carol Davila” University of Medicine and Pharmacy, 050098 Bucharest, Romania; 3Internal Medicine Department, Emergency University Hospital of Bucharest, “Carol Davila” University of Medicine and Pharmacy, 050098 Bucharest, Romania; cora.pop@gmail.com

**Keywords:** liver, T2DM, COVID-19, FIB-4

## Abstract

*Background:* Emerging evidence suggests that patients with metabolic (dysfunction) associated fatty liver disease (MAFLD) are prone to severe forms of coronavirus disease (COVID-19), especially those with underlying liver fibrosis. The aim of our study is to assess the association of an increased FIB-4 score with COVID-19 disease prognosis. *Methods:* We performed a prospective study on hospitalized patients with known type II diabetes mellitus (T2DM) and confirmed COVID-19, with imaging evidence of liver steatosis within the last year or known diagnosis of MAFLD. All individuals were screened for liver fibrosis with a FIB-4 index. We evaluated the link between FIB-4 and disease prognosis. *Results:* Of 138 participants, 91.3% had MAFLD and 21.5% patients had a high risk of fibrosis. In the latter group of patients, the number of severe forms of disease, the hospital stay length, the rate of ICU admissions and the number of deaths reported registered a statistically significant increase. The independent predictors for developing severe forms of COVID-19 were obesity (odds ratio (OR), 3.24; 95% confidence interval (CI), *p* = 0.003), higher values of ferritin (OR-1.9; 95% CI, 1.17–8.29, *p* = 0.031) and of FIB-4 ≥ 3.25 (OR-4.89; 95% CI, 1.34–12.3, *p* = 0.02). *Conclusions:* Patients with high scores of FIB-4 have poor clinical outcomes and liver fibrosis may have a relevant prognostic role. Although the link between liver fibrosis and the prognosis of COVD-19 needs to be evaluated in further studies, screening for liver fibrosis with FIB-4 index, particularly in patients at risk, such as those with T2DM, will make a huge contribution to patient risk stratification.

## 1. Introduction

Coronavirus disease (COVID-19) is an infectious condition caused by a newly identified member of the coronavirus family, the severe acute respiratory syndrome coronavirus 2 (SARS-CoV-2), which causes predominant pulmonary damage and may lead to acute respiratory distress syndrome (ARDS) and death [1]. Due to its rapidly expanding and sustained risk of global spread, The World Health Organization (WHO) declared COVID-19 a pandemic on 11 March 2020 [2]. Since then, scientific efforts around the world have been made to establish risk factors for the development of a severe form of disease and death.

According to The Center for Disease Control and Prevention (CDC) that described and updated the comorbidities linked to a poor prognosis in COVID-19, metabolic disorders, such as type II diabetes mellitus (T2DM) and obesity ((BMI) > 30 kg/m^2^), increase the risk for severe illness and death [3]. Moreover, this constellation of metabolic derangements is considered to be a risk factor for the development of metabolic (dysfunction) associated fatty liver disease (MAFLD), a consensus-driven acronym for non-alcoholic fatty liver disease (NAFLD) [4,5]. MAFLD, which is the hepatic consequence of metabolic syndrome, includes a wide clinical spectrum, from simple steatosis (SS), non-alcoholic steatohepatitis (NASH), to cirrhosis and MAFLD-related hepatocellular carcinoma (HCC) [6]. With a global prevalence of 25%, MAFLD is regarded as the leading cause of advanced liver disease, especially among those with T2DM [7].

Globally, there is growing evidence that patients with MAFLD are prone to severe forms of COVID-19, as they feature a combination of metabolic comorbidities, such as T2DM and obesity [8,9]. MAFLD regularly coexists with T2DM: the presence of MAFLD increases the incidence of T2DM, while T2DM accelerates the development of NASH, cirrhosis and even NAFLD-related HCC [10]. In a recently published systematic review carried out by our team, we described the inter-relationship between MAFLD, T2DM and COVID-19. Patients with MAFLD presented severe forms of SARS-CoV-2 infection, had longer viral shredding time and a higher probability for developing abnormal liver function tests (LFT) from admission to discharge [11]. Numerous interconnected metabolic and inflammatory pathophysiologic consequences link T2DM and MAFLD to COVID-19 severity; MAFLD causes insulin resistance with dysfunctional fatty acid metabolism, generating a state of low-grade inflammation that seems to impair immune responses that contribute to a greater chance of COVID-19 “cytokine storm” [12]. Furthermore, underlying T2DM is associated with immunological dysregulation with abnormal cytokine response that may exacerbate the hyperinflammatory response in individuals with SARS-CoV-2 infection [13].

Early studies report elevations in liver biochemistry in up to 85% of patients with MAFLD and confirmed COVID-19, with higher prevalence in those with underlying liver fibrosis [14,15]. An extensive meta-analytic systematic review of seven studies carried out between 2004 and 2017 including 439 liver biopsies of patients with T2DM showed that the mean prevalence of reported severe fibrosis was 22.01% (range: 3.39% to 50.00%) [16]. A common pathway in the pathophysiology of metabolic conditions, including T2DM and MAFLD, is the presence of a low-grade chronic inflammatory status that contributes to the development of liver fibrosis, and, further on, can exacerbate the COVID-19-induced “cytokine storm” [17]. Moreover, the respiratory infection rate is higher in cirrhotic patients, with respiratory viruses being detected in up to 22.2% of these patients as compared to non-cirrhotic ones [18].

Overall, there still is insufficient information as to how liver fibrosis in diabetic populations impacts the progression of SARS-CoV-2 infection. Given that liver biopsy is not frequently accepted and it has no benefit in diabetic patients with COVID-19 for MAFLD screening, non-invasive tests using serum biomarkers are more appropriate for this subset of patients. Therefore, the aim of this study is to evaluate the association of high FIB-4 index in patients with T2DM, with COVID-19 disease prognosis [19].

## 2. Materials and Methods

### 2.1. Study Population

We performed a prospective, descriptive study in Bucharest Emergency University Hospital, Romania, according to the guidelines of the Declaration of Helsinki, and approved by the Ethics Committee of Emergency University Hospital of Bucharest (no. 9195/17.02.2021). Inclusion criteria were adult patients (>18 years of age) with known T2DM, diagnosed with COVID-19 between October 2020–February 2021. All patients were evaluated for liver steatosis by checking the previous year’s medical history (abdominal ultrasound/biochemical enzymes) or by computer tomography scans of the thorax performed upon admission for lung damage evaluation which included images at the level of the upper pole of the liver and spleen. T2DM was defined based on the criteria of the American Diabetes Association (ADA) “Standards of Medical Care in Diabetes” [20]. SARS-CoV-2 infection was detected in nasopharyngeal swabs using a time reverse transcriptase-polymerase chain reaction (RT-PCR). Patients with other etiologies of liver diseases including autoimmune hepatitis, drug-related hepatitis, chronic hepatitis C, chronic hepatitis B, Wilson’s disease, haemochromatosis and conditions such as myopathies and platelet disorders were excluded. Similarly, patients with alcohol consumption >20 g/day for females and >30 g/day for males were excluded from the analysis. We enrolled a total of 138 diabetic patients with confirmed SARS-CoV-2 infection. Informed consent was obtained from all subjects involved in the study.

### 2.2. Clinical Examination and Laboratory Tests

At the time of presentation, the demographic data, clinical and laboratory parameters were collected for each subject. The body mass index (BMI) was defined as body weight (kilograms) divided by the square of body height (m^2^). All patients were interrogated for comorbidities, alcohol consumption and drug history.

COVID-19 was diagnosed according to hospital protocols and recent WHO guidelines [21]. Clinical parameters such as arterial blood pressure, heart rate (HR), respiratory rate (RR), temperature, saturation and a history of pre-admission symptoms were recorded. Lung damage was assessed by chest radiographs or computer tomography scans. SARS-CoV-2 infection severity was classified according to the “Diagnosis and Treatment Protocol for Novel Coronavirus Pneumonia” as severe and non-severe [22]. Severe cases were defined by the presence of any of the following criteria: (1) respiratory distress (≥30 breaths/min); (2) oxygen saturation ≤ 93% at rest; (3) arterial partial pressure of oxygen (PaO_2_)/fraction of inspired oxygen (FiO_2_) ≤ 300 mmHg. Mild and moderate cases were pooled into a non-severe group. Blood tests were performed for all patients, including complete blood count (CBC), liver function tests (Aspartate Aminotransferase (AST), Alanine Aminotransferase (ALT), Gamma-glutamyltransferase (GGT), Alkaline Phosphatase (ALP), Lactate dehydrogenase (LDH), total bilirubin, direct bilirubin and albumin), total cholesterol, triglycerides, C-reactive protein (CRP), ferritin, fasting glucose and glycosylated hemoglobin (HbA1C).

### 2.3. Liver Steatosis and Fibrosis Assessment 

NAFLD was diagnosed by checking all patients’ history for imaging evidence (ultrasound or computer tomography) and/or biochemical enzymes (LFT) within the past 12 months. Patients with no history of liver steatosis were screened for NAFLD via non-contrast chest CT which included images from the liver at the level of right portal branch and the upper pole of the spleen. A single radiologist with high expertise analyzed the CT scans for the presence of liver steatosis using the relative hypoattenuation and absolute low attenuation criteria [23]. Out of the 138 patients enrolled in this study, 126 (91.3%) had NAFLD. All individuals with NAFLD were screened for liver fibrosis by FIB-4 index, which was calculated with the formula: (Age*AST (IU/L))/(Platelets (×109/L) * sqrt(ALT (IU/L))). [24] We distinguished three groups of patients according to prior validated cut points: patients with low risk of fibrosis (FIB-4 < 1.30), patients with intermediate risk of fibrosis (FIB-4: 1.30–3.25), patients with high risk of fibrosis (FIB-4 > 3.25) [24]. 

### 2.4. Statistical Analysis

All data were stored in Microsoft Office Excel, 2019 version. The statistical analysis was performed using Epi Info version 7.2.4.2020. Continuous variables were presented as means ± standard deviation (SD) and the non-continuous ones were expressed in percentages or frequencies. The distribution of variables was evaluated by the Kolmogorov–Smirnov test. Analysis of Variance (ANOVA) was used to determine whether there is a statistically significant difference between data sets. Multivariate logistic regression analysis was used to create a regression model which has been modified to take into account other predictor variables. The results were reported as adjusted odds ratios (AOR). A value of *p* < 0.005 was considered significant.

## 3. Results

### 3.1. Baseline Characteristics of Enrolled Patients

In our study population of 138 diabetic patients with COVID-19, 91.3% (*n* = 126) had NAFLD, while 8.7% (12) had no evidence of liver disease. The baseline characteristics of our cohort are shown in Table 1. The mean age was 66.32 ± 13.72 with a higher proportion of men (57.2% (*n* = 79)). The mean BMI was 29.91 ± 5.28. Overall, the most frequent comorbidity was obesity (48.5%), followed by hypertension (46.2%) and dyslipidemia (10.2%). No active malignancy was registered. Patients had a median of 5 (3–9) days between the onset of symptoms and the hospital presentation. Symptoms included cough (69.3%), dyspnea (58.2%) and headache (37.2%). The most frequent gastrointestinal (GI) symptom reported was diarrhea (34.9%), followed by nausea (13.1%) and abdominal pain (10.2%). Regarding COVID-19 severity, 36.3% (*n* = 50) patients were classified as non-severe and 63.7% (88) patients as severe. The median in-hospital stay was 16 (11–22) days. A 44.9% (*n* = 62) rate of ICU admission and 18.1% (*n* = 25) deaths were reported. Among laboratory tests, the median lymphocytes were 9.7% (6.3–11.2), platelets 186.2 cell × 10^3^ U/L (123.4–224.5), leukocytes 8.2 cell × 10^3^ U/L (6.3–11.11), while CRP was 37.1 mg/dL ± 2.33, ferritin 677.4 ng/mL ± 221.3, serum glucose 179.35 mg/dL ± 85.2 and HbA1C 6.76% ± 1.12. The prevalence of MAFLD was 91.3% (*n* = 126), while 8.7% (*n* = 12) had no evidence of liver steatosis. Regarding the FIB-4 values, patients with MAFLD were stratified into three groups: 62.7% (*n* = 79) patients with low risk of fibrosis (FIB-4 < 1.30), 15.8% (*n* = 20) patients with intermediate risk of fibrosis (FIB-4: 1.30–3.25) and 21.5% (27) patients with high-risk fibrosis (FIB-4 > 3.25).

LFT showed that 73.9% (*n* = 102) patients had at least one abnormal parameter upon hospital admission, and lactate dehydrogenase (LDH) was the most frequent (81.5%, *n* = 106). The injury pattern was mostly hepatocellular (64.6%, *n* = 84), while cholestatic liver enzymes were increased in the later stages of the disease. The median values of LFT are shown in Table 2. 

### 3.2. Differences between Patients Stratified by Fibrosis Status

The comparison between the clinical and biochemical parameters of the three groups stratified by FIB-4 values is shown in Table 3. Patients with NAFLD and intermediate and high-risk of fibrosis were older (68.5 ± 12.2 vs. 64.25 ± 11.21 vs. 61.3 ± 10.5, *p* < 0.001) and had higher BMI (31.3 ± 5.6 vs. 27.9 ± 5.2 vs. 27.2 ± 4.5, *p* < 0.001). Moreover, patients in the high-risk fibrosis group had fewer days between the onset of symptoms and the hospital presentation (4 (2–6) vs. 6 (3–9) vs. 6 (3–9), *p* = 0.034). As compared to other groups of patients, patients with NAFLD and high FIB-4 index scores were more likely to have higher values of ferritin (690.57 ng/mL ± 197.85 vs. 625.87 ng/mL ± 201.24 vs. 623.45 ng/mL ± 198.56, *p* = 0.013), serum glucose (198.25 mg/dL ± 87.68 vs. 164.54 mg/dL ± 64.23 vs. 156.78 mg/dL ± 65.24, *p* < 0.001), HbA1C (7.8% (6.7–9.1) vs. 7.2% (6.6–8.4) vs. 7.1% (6.4–8.3), *p* = 0.037) and higher LFT. Regarding the features of systemic inflammatory response, lymphocytes (8.3% vs. 8.70% vs. 9.5%, *p* = 0.095) were lower in the high-risk fibrosis group and C-reactive protein (38.45 mg/dL ± 3.45 vs. 37.23 mg/dL ± 4.89 vs. 37.76 mg/dL ± 4.56, *p* = 0.243) was slightly increased, but this was not statically significant.

In the group of patients with high-risk of fibrosis the hospital stay was statistically significantly longer (17 days vs. 13 days vs. 11 days, *p* < 0.014), as well as the rate of ICU admission (23% vs. 17.4% vs. 8.7%, *p* = 0.021) and number of deaths reported (10.3% vs. 6.3% vs. 3.1%, *p* < 0.001). The severity of SARS-CoV-2 infection increased among patients with intermediate and high risk of fibrosis (38.8% vs. 21.4% vs. 9.5%, *p* < 0.001).

### 3.3. Factors Associated with Poor Prognosis in NAFLD Population

Multivariate logistic regression identified the risk factors associated with developing a severe form of COVID-19 *(*Table 4*)*. The independent predictors were obesity (OR-3.24; 95% CI, 1.46–5.32, *p* = 0.003), higher values of ferritin (OR-1.9; 95% CI 1.78-8.29, *p* = 0.031) and FIB-4 index scores (OR-4.89; 95% CI, 1.34-12.3, *p* = 0.02).

## 4. Discussion

Advanced chronic liver disease in patients with T2DM mostly occurs due to NAFLD and is expected to become the first cause of hepatic transplantation [25]. While the global pandemic of COVID-19 evolves, emerging research highlights the crucial role of liver fibrosis in SARS-CoV-2 infection severity [26]. In the NAFLD clinical spectrum, the recognition of patients at high-risk for liver fibrosis should be the primary goal, because several large studies have found that liver fibrosis is a strong independent predictor for mortality, liver transplantation and NAFLD-associated hepatocellular carcinoma (HCC) [27]. As the evidence on the link between these comorbidities and the prognosis of COVID-19 is still scarce, our study aimed at evaluating the association between a non-invasive liver fibrosis score and the risk of progression to severe forms of the disease. Given that liver biopsy is not frequently accepted and generates no benefits in diabetic patients with COVID-19 for NAFLD screening, we selected FIB-4 index, a non-invasive test to assess liver fibrosis. Sterling et al. initially created the FIB-4 index for evaluating severe liver fibrosis in patients with hepatitis C virus (HCV) and human immunodeficiency virus (HIV) [28,29]. According to the American Association of Liver Disease, who recently published a guideline for NAFLD management, FIB-4 score was the most clinically effective index for assessing liver fibrosis in patients with NAFLD. In this guideline, FIB-4 was compared to other scores like AST to Platelet Ratio Index (APRI), NAFLD fibrosis score (NFS), AST/ALT ratio and BARD [19]. Moreover, its performance was validated in a large study in Japan, where FIB-4 index was superior to other non-invasive tests, with a high negative predictive value for excluding advanced hepatic fibrosis [30].

In our study population of 138 patients with T2DM, admitted for COVID-19, males outnumbered females by 1.3:1. The prevalence of NAFLD was 91.3% (*n* = 126), from which 21.5% (*n* = 27) patients were at high-risk for liver fibrosis according to the FIB-4 index. A metanalysis by Lonardo et al., which consisted of 28 longitudinal studies, showed that the prevalence rate of NAFLD in diabetic populations varied from 50% to 78%, according to ethnicity [31]. Rocío Aller de la Fuente et al. found in a study that analyzed 217 liver biopsies from a Western population with known T2DM that 80.6% had steatosis [32]. There is a high variation in research regarding the prevalence of NAFLD in diabetic populations depending on the diagnostic test that was used [33]. We want to emphasize that the reason that we found a high rate of NAFLD in our study cohort is because obesity was the most prevalent comorbidity. As studies all over the world showed, there is a dose-response effect between obesity and severe forms of COVID-19, with obese patients frequently requiring hospitalization [34,35]. The prevalence of advanced liver fibrosis in our cohort, as stratified by FIB-4 index, was consistent with a multicenter study in Spain, which included 160 patients and reported a 28.1% rate of advanced liver fibrosis [36]. Previous studies that used liver biopsy as gold standard have found that the prevalence of severe fibrosis among T2DM patients ranges from 16.2–43.1%, which is similar to our report [32,37]. 

The pathogenesis of liver involvement in COVID-19 is likely multifactorial, including direct cytopathic effect of the virus, liver hypoxia and ischemia, sepsis or uncontrolled immune reaction. Additionally, several reports showed that SARS-CoV-2 receptor, the angiotensin-converting enzyme 2 (ACE2), has an increased expression on cholangiocytes and a lower expression in hepatocytes, explaining why the virus only causes mild and transient hepatic damage [38,39]. We found a 73.9% abnormal LFT at hospital presentation in our patients. The injury pattern was mostly hepatocellular (64.6%, *n* = 84), while cholestatic liver enzymes were increased in the later stages of the disease, which is consistent with other studies that evaluated liver involvement in NAFLD patients due to COVID-19 [26,40]. While higher values of ALP and GGT are considered rather rare at hospital admission [41], a recent metanalysis describes elevations of ALP and GGT during hospitalization in 6.1% and 21.1% of COVID-19 patients, respectively [42]. This dual response with initial AST and ALT elevation followed by ALP and GGT has been first reported in Chinese cohorts, which could suggest bile duct damage induced by Systemic Inflammatory Response Syndrome (SIRS) in the later stage of disease [39,43]. Another study on 202 patients with NAFLD and confirmed COVID-19 found that almost all liver involvement was mild with hepatocellular pattern, a result similar to ours [26].

The most important discovery in our report was the link between the high-risk fibrosis scores and severe forms of COVID-19. In the group of patients with high FIB-4 scores the hospital stay was statistically significant increased (17 days vs.13 days vs. 11 days, *p* = 0.014), as well as the rate of ICU admission (23% vs. 17.4% vs. 8.7%, *p* = 0.021) and number of deaths reported (10.3% vs. 6.3% vs. 3.1%%, *p* < 0.001). Also, the severity of SARS-CoV-2 infection increased among patients with intermediate and high risk of fibrosis (38.8% vs. 21.4% vs.9.5%, *p* < 0.001). Similar results have been reported by Ibáñez-Samaniego et al. who showed, in a multicenter, retrospective study, that patients with high scores of FIB-4 have poor clinical outcomes and liver fibrosis may have a relevant prognostic role [36]. In this context, we can hypothesize that advanced liver fibrosis may promote a dysfunctional inflammatory response that characterizes SARS-CoV-2 infection. Severe liver fibrosis associates an immune dysfunction that is the consequence of two interlinked mechanisms, systemic inflammation and damage of the immune system response. This immune dysfunction is characterized by a persistent immune cells activation by damage-associated molecular patterns (DAMPs) from damaged tissues, especially necrotic hepatocytes. Moreover, pathogen-associated molecular patterns (PAMPs) released from the leaky gut further upregulate the immune system and the production of proinflammatory cytokines which recruit additional inflammatory cells, creating a state of low-grade systemic inflammation [44,45]. This process is exacerbated by the presence of other metabolic derangements, such as obesity, NAFLD, insulin-resistance or T2DM, comorbidities with a high prevalence in our study population [46]. Therefore, we found in our cohort that inflammatory response is stronger in patients with higher scores of FIB-4, as suggested by elevation of acute-phase response proteins, such as CRP and ferritin. 

Furthermore, when looking at the independent effect of the researched variables on the clinical outcome of the disease, multivariate logistic regression found that the presence of obesity (OR-3.24, *p* = 0.003), and the higher values of ferritin (OR-1.9, *p* = 0.031) and FIB-4 index (OR-4.89, *p* = 0.002) increase the chances of developing a severe form of COVID-19. Similar results have been found by Targher G. et al. in one of the few existing studies in scientific research that evaluates the role of liver fibrosis in COVID-19 clinical outcomes [47]. A high number of reports have showed obesity as a risk factor for severe SARS-COV2 infection. The OpenSafely study which included over 17 million participants and analyzed mortality in hospitals among people with confirmed SARS-CoV-2 infection, showed increasing risk of death with obesity stage: from a risk of 27% in the first class of obesity (BMI 30–34.9; hazard ratio 1.27, 1.18 to 1.36) to more than double in the third class (BMI > 40; 2.27, 1.99 to 2.58) [34]. Regarding the symptoms at presentation, a systemic literature review which included thirteen studies with a total number of 3027 patients with confirmed COVID-19 and analyzed risk factors of critical/non-critical form of disease, found that shortness of breath was associated with the progression of disease [48]. Our results are consistent, dyspnea being the only predictive symptom for developing a severe form of COVID-19 (OR-2.19, *p* = 0.042). 

This study has some limitations. Our cohort included a diabetic population of Caucasians which showed a high prevalence of obesity and advanced liver fibrosis. In addition, the cohort is a small one, and larger studies need to be validated on different ethnicities. Another limitation concerns FIB-4 score. First, constituents of the FIB-4 index (AST, ALT and platelets) may be influenced by disorders other than hepatic diseases. For this, myopathies and platelet disorders were exclusion criteria for our study population. Secondly, given that elevation of liver enzymes is widely reported in SARS-CoV-2 infection, [42,49] FIB-4 may by falsely raised and not necessarily a precise test for evaluating liver fibrosis. Therefore, we tried to raise the efficacy of the index by retrieving available blood tests from last year for each patient. A small number of patients (8%) from whom we did not find information about liver enzymes from previous tests showed elevated AST and ALT at the time of hospitalization for COVID-19, but their thrombocytes remained the same compared with previous values. Yet, there were no notable changes in their FIB-4 index stratification. Several studies that used the FIB-4 index to assess liver fibrosis in patients with COVID-19 have overcome this limitation using a data analysis similar to ours [36]. Thirdly, FIB-4 score may perform poorly for diagnosing low and intermediate risk of fibrosis in patients at the extremes of age (<35 years and >65 years). However, the cutoff for advanced fibrosis remains the same in all age groups [50]. Another limitation is that liver fibrosis was diagnosed by a non-invasive score which is not the gold standard method. However, given the specific risk for SARS-CoV-2 transmission among healthcare workers, the use of a non-invasive score like FIB-4 index was safer in order to prevent exposure. Finally, we want to highlight that assessing the presence of liver fibrosis using a method different from biochemical markers is very difficult to perform and it involves epidemiological risks during the COVID-19 pandemic. Despite these limitations, based on our literature research, our study is one of the few reports which evaluates the impact of liver fibrosis in a population at risk, such as those with T2DM, on SARS-CoV-2 infection clinical outcomes. 

## 5. Conclusions

To conclude, our study analyzed and reported the impact of high FIB-4 index in a diabetic population on COVID-19 disease prognosis. In the group of patients with high FIB-4 scores, the hospital stay was statistically significant increased, as well as the number of severe forms of disease, the rate of ICU admission and number of death reports. Even if non-invasive tests using biochemical markers may be affected by the SARS-CoV-2 infection and the link between liver fibrosis and COVID-19 clinical outcome needs to be clarified in further studies, we promote screening with FIB-4 score, particularly in patients at risk, such as those with T2DM. We believe that this will make a huge contribution to patient risk stratification and will be a great help to healthcare workers to decrease the associated health care cost, especially in regions where metabolic disorders have a high prevalence.

## Figures and Tables

**Table 1 medicina-57-00434-t001:** Clinical and biochemical characteristics of the study population (*n*-138).

Male gender (%)	57.2% (79)
Age	66.32 ± 13.72
BMI	29.91 ± 5.28
COMORBIDITIES	
Cardiovascular diseases	5.80%
Hypertension	46.20%
Obesity	48.50%
Dyslipidemia	10.20%
Pulmonary disease	3.80%
Neurologic Disease	4.10%
Kidney Diseases	1.40%
Smoking	32.30%
SYMPTOMS AT PRESENTATIONS	
Symptoms before presentation (days)	5 (3–9)
Headache	37.2%
Dyspnea	58.2%
Fever	31.2%
Cough	69.3%
Chest pain	22.4%
Abdominal pain	10.2%
Diarrhea	34.9%
Nausea	13.1%
Vomiting	5.2%
Anosmia	6.8%
Dysgeusia	4.7%
CLINICAL EXAMINATION AT PRESENTATION	
Systolic BP (mmHg)	126 (111–149)
Heart Rate (bpm)	94 (84–108)
Saturation (%)	88 (82–91)
Respiratory Rate (rpm)	24 (22–28)
HOSPITAL EVOLUTION	
Non-severe form of COVID-19 (%)	36.3% (50)
Severe form of COVID-19 (%)	63.7% (88)
Hospital stay (days)	16 (11–22)
ICU admission (%)	44.9% (62)
Deaths (%)	18.1% (25)
LABORATORY TESTS	
Leukocytes (cell × 10^3^ U/L)	8.2 (6.3–11.11)
Lymphocytes (%)	9.7 (6.3–11.2)
Platelets (cell × 10^3^ U/L)	186.2 (123.4–224.5)
C-reactive protein (mg/dl)	37.1 ± 2.33
Ferritin (ng/mL)	677.4 ± 221.3
Total cholesterol (mg/dL)	144.55 ± 49.29
Serum glucose (mg/dL)	179.35 ± 85.2
HbA1C (%)	6.76 ± 1.12
LIVER STEATOSIS AND FIBROSIS ASSESSMENT	
Liver steatosis (NAFLD)	91.3% (126)
FIB-4 < 1.30	62.7% (79)
FIB-4 1.30–3.25	15.8% (20)
FIB-4 > 3.25	21.5% (27)

BMI—Body mass index; BP—blood pressure; ICU—intensive care unit; HbA1C—glycosylated hemoglobin.

**Table 2 medicina-57-00434-t002:** Liver function tests of the study population (*n*-138).

AST (U/L)	68.47 (49.2–80.7)
ALT(U/L)	58.7 (42.4–73.6)
GGT (U/L)	87.2 (39.7–109.3)
ALP (U/L)	77.9 (53.4–93.4)
LDH (U/L)	325 (248–384)
Total bilirubin (mg/dL)	1.2 (0.84–1.3)
Direct bilirubin (mg/dL)	0.4 (0.26–0.53)
Albumin (g/dL)	3.2 (2.9–3.7)

AST—aspartate amino transferase; ALT—alanine amino transferase; GGT—gamma-glutamyltransferase; ALP—alkaline phosphatase; LDH—lactate dehydrogenase.

**Table 3 medicina-57-00434-t003:** Difference between patients with NAFLD stratified by fibrosis status (FIB-4) (*n*-126).

	FIB-4 < 1.3 (79)	FIB-4 1.3–3.25 (20)	FIB-4 > 3.25 (27)	*p* Value *
Age (years)	61.3 ± 10.5	64.25 ± 11.21	68.5 ± 12.2	<0.001
BMI	27.2 ± 4.5	27.9 ± 5.2	31.3 ± 5.6	<0.001
Symptoms before presentation	6 (3–9)	6 (3–9)	4 (2–6)	0.034
Lymphocytes (%)	9.5%	8.70%	8.3%	0.095
C-reactive protein (mg/dL)	37.76 ± 4.56	37.23 ± 4.89	38.45 ± 3.45	0.243
Ferritin (ng/dL)	623.45 ± 198.56	625.87 ± 201.24	690.57 ± 197.85	0.013
Serum glucose (mg/dL)	156.78 ± 65.24	164.54 ± 64.23	198.25 ± 87.68	<0.001
HbA1C (%)	7.1 (6.4–8.3)	7.2 (6.6–8.4)	7.8 (6.7–9.1)	0.037
AST (U/L)	44 (29–85)	49 (36–75)	57 (37–88)	<0.001
ALT (U/L)	35 (26–58)	37 (21–62)	47 (29–76)	<0.001
GGT (U/L)	69 (41–102)	72 (45–99)	94 (45–123)	<0.001
ALP (U/L)	53 (31–81)	62 (42–91)	79 (49–98)	<0.001
LDH (U/L)	317 (198–349)	345 (215–357)	391 (315–434)	0.002
Total bilirubin (mg/dL)	0.9 (0.6–1.1)	0.8 (0.5–1.2)	1.1 (0.7–1.5)	0.067
Albumin (g/dL)	3.8 (2.8–4.2)	3.9 (3.0–4.5)	3.7 (2.5–4.3)	0.214
Total cholesterol (mmol/L)	142.45 ± 47.25	140.57 ± 49.58	143.37 ± 48.56	0.458
Hospital stay (days)	11 (7–15)	13 (9–17)	17 (12–21)	0.014
ICU admission (%)	8.7% (11)	17.4% (22)	23% (29)	0.021
Severe form of COVID-19 (%)	9.5% (12)	21.4% (27)	38.8% (49)	<0.001
Deaths (%)	3.1% (4)	6.3% (8)	10.3% (13)	<0.001

BMI—Body mass index; HbA1C—glycosylated hemoglobin; AST—aspartate amino transferase; ALT—alanine amino transferase; GGT—gamma-glutamyltransferase; ALP—alkaline phosphatase; LDH—lactate dehydrogenase; ICU—intensive care unit. * Values are statistically significant at *p* < 0.05.

**Table 4 medicina-57-00434-t004:** Risk factors associated with severe form of COVID-19.

	Adjusted OR * (95% CI)	*p* Value **
Male gender (male vs. female)	1.28 (0.92–3.53)	0.587
Obesity (yes/no)	3.24 (1.46–5.32)	0.003
Dyspnea (yes/no)	2.19 (1.56–6.29)	0.042
C-reactive protein (every 10mg/L increment)	0.56 (1.021–1.045)	0.654
Ferritin (every 100 ng/mL increment)	1.9 (1.78–8.29)	0.031
AST (every 10 U/L increment)	1.67 (0.94–1.03)	0.087
ALT (every 10U/L increment)	0.87 (0.99–1.09)	0.354
FIB-4 < 1.3	1.25 (0.40–2.45)	0.847
FIB-4 1.3–3.25	2.47 (1.01–7.63)	0.057
FIB-4 > 3.25	4.89 (1.34–12.3)	0.002

AST—aspartate amino transferase; ALT—alanine amino transferase. Obesity was diagnosed as BMI > 25 kg/m^2^. * the OR was adjusted for sex, BMI, the presence of dyspnea and the levels of ferritin, CRP, AST, ALT and FIB-4 score. We did not additionally adjust for age, because this variable in already incorporated in the FIB-4 score. ** Values are statistically significant at *p* < 0.05.

## Data Availability

Additional data that support the findings of this study are available from the corresponding author [L.C.T.] and [M.O.], upon reasonable request.

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
