# Peer review of "The Impact of Increased Fib-4 Score in Patients with Type II Diabetes Mellitus on COVID-19 Disease Prognosis"

_medicina, 2021, doi:10.3390/medicina57050434_

Round 1
Reviewer 1 Report
They have addressed my comments
Author Response
RESPONSE FOR REVIEWER 1
Comments and Suggestions for Authors: “They have addressed my comments.”
Response: I am deeply grateful for your comments on our paper. Thank you for the time and effort that you have dedicated to providing your valuable feedback on our manuscript.
Kind Regards,
Laura Tribus
Reviewer 2 Report
Executive Summary
The revised form of the manuscript titled “The impact of increased FIB-4 score in patients with type II diabetes mellitus on Covid-19 disease prognosis” conducted clinical research investigating the correlation between FIB-4 value and Covid-19 related characteristics. Overall, the revision focuses on FIB-4 rather than liver fibrosis, based on previous peer review. Authors may perform minor revisions to further improve the quality of this manuscript.
Major Comments
There is no major comment.
Minor Comments
- Authors can still claim that Type II diabetic patients with a high FIB-4 index may have a higher chance to have the severe form of Covid-19 based on data from Table 4 (P-value: 0.057 and 0.002), and data from Table 3 (P-value <0.001). This was not claimed in the abstract or conclusion.
- Authors may revise the abstract accordingly.
- Authors may revise the conclusion section accordingly.
Author Response
RESPONSE FOR REVIEWER 2
I am deeply grateful for your comments on our paper. Thank you for the time and effort that you have dedicated to providing your valuable feedback on our manuscript. I have clearly highlighted the changes within the manuscript using the “Track changes” function in Microsoft Word.
Comments and Suggestions for Authors:
“Executive Summary
The revised form of the manuscript titled “The impact of increased FIB-4 score in patients with type II diabetes mellitus on Covid-19 disease prognosis” conducted clinical research investigating the correlation between FIB-4 value and Covid-19 related characteristics. Overall, the revision focuses on FIB-4 rather than liver fibrosis, based on previous peer review. Authors may perform minor revisions to further improve the quality of this manuscript.
Major Comments
There is no major comment.
Minor Comments
- Authors can still claim that Type II diabetic patients with a high FIB-4 index may have a higher chance to have the severe form of Covid-19 based on data from Table 4 (P-value: 0.057 and 0.002), and data from Table 3 (P-value <0.001). This was not claimed in the abstract or conclusion.
- Authors may revise the abstract accordingly.
- Authors may revise the conclusion section accordingly.”
Response: Thank you for your suggestions. We added this in the abstract at line number 28-29, and in conclusions section at line number 369.
Kind Regards,
Laura Tribus
This manuscript is a resubmission of an earlier submission. The following is a list of the peer review reports and author responses from that submission.
Round 1
Reviewer 1 Report
The authors aimed to assess the association of liver fibrosis with COVID-19 clinical outcomes in their study. This is an original research of high interest to the readers. The introduction of the paper provides sufficient background and all relevant references. Although the transversal design of the study is appropriate for this research, the methods are not adequate enough described and the results are not clear. Also, moderate English changes are required. I have the following comments:
- the abstract: there are 2 similar ideas, please consider rewriting the aim of the study without repeating the idea " The aim of our study is to assess the association of liver fibrosis with COVID-19 clinical outcomes. .... We evaluated the link between FIB-4 and disease prognosis. ". The conclusion from the abstract should be modified. "The prevalence of liver fibrosis is high in patients with T2DM" the prevalence of liver fibrosis was not an outcome in this study, therefore it should not be there. Please consider writing a specific conclusion of the study and its applicability, avoiding general statements.
- it is not very clear what was the final study group, 126 patients with NAFLD which were screened for the FIB-4 index? Because the first table analyzed all the recruited patients. Please mention the number of patients analyzed in each table.
- also, it is not clear how the patients were diagnosed with NAFLD by CT scan, please describe the whole process in detail.
- The Statistical analysis is very averagely described. Was the data tested for normal distribution? To compare the data from this research, the t-student test is not always adequate. Also, the multivariate logistic regression model is not very clearly presented. Please consider revising the whole statistical analysis, in order to enhance the clarity and validity of this research.
- please add the measurement units for the variables in the whole text
- on what severity criteria were the COVID19 patients classified?
- how many deaths were in each group, of the total deaths? to gain clarity another test should be used here. Please revise table 3 in regards to COVID19 prognosis, comparing the groups between them, out of the total number of patients.
- table 4 if not very clear, please describe in the footnote what type of variables are each. Also, consider revising this analysis.
- the discussions must be more organized and focused on the main aim of the present study, the association between liver fibrosis and COVID19 prognosis, comparing these results with similar studies, then suggesting a possible mechanism underlying these results.
- and last, please revise the conclusion, be concise and clear, avoiding general statements. " advanced liver fibrosis associated with either T2DM or obesity is an independent predictor for the prognosis of patients with NAFLD and SARS-COV-2 infection" is not the conclusion of the study. Liver fibrosis doesn't predict SARS-COV-2 infection.
Author Response
Please see the attachment.
Kind regards,
Laura Tribus

Reviewer 2 Report
Authors have shown an expected result of higher morbidity and mortality among T2DM patients with high risk of cirrhosis.
The study conducted by Calapod et.al. aimed at evaluating the outcomes of type 2 diabetes patients with advanced liver fibrosis and infected with COVID-19. the authors prospectively included 138 individuals and compared the outcomes based on fibrosis score. They found that patients with fibrosis score more than 3.25 heard higher percentage of patients with need for ICU admission, her longer length of hospital stay, had higher severity of COVID-19 infection as well as mortality. These results have been shown previously but nevertheless are important to be shown among different set of population for validity.
Author Response
Please see the attachment.
Kind Regards,
Laura Tribus

Reviewer 3 Report
The main objective of the study was to assess if patients with T2DM and advanced liver fibrosis had worse COVID-19 outcomes. Unfortunately, while the idea behind the study was interesting, there is a conceptual mistake in the design:
The authors used FIB-4 as a surrogate marker of liver fibrosis, a parameter that is calculated based on AST, ALT, platelets and age. Irrespective of the amount of liver fibrosis, severe COVID-19 infections are known to produce elevations in AST/ALT as well as thrombocytopenia. Therefore, in the setting of COVID-19 infection this score cannot be used to predict liver fibrosis, as it is being affected by the COVID-19 infection.
Moreover, because severe infections are likely to modify this score more pronouncedly, the associations reported cannot be attributed to the severity of liver fibrosis.
Author Response

(The authors gave the same response as above.)

Round 2
Reviewer 1 Report
The new version of the manuscript is significantly improved.